

# Body size information in large-scale acoustic bat databases

Caterina Penone[1], Christian Kerbiriou[2,3], Jean-François Julien[4], Julie Marmet[4] and Isabelle Le Viol[3,4]

[1] Institute of Plant Sciences, University of Bern, Bern, Switzerland
[2] CESCO UMR7204 MNHN-UPMC-CNRS-Sorbonne Université, Université Pierre et Marie Curie (Paris VI), Paris, France
[3] Marine Station, Muséum national d'Histoire naturelle, Concarneau, France
[4] CESCO UMR7204 MNHN-UPMC-CNRS-Sorbonne Université, Muséum national d'Histoire naturelle, Paris, France

## ABSTRACT

**Background**. Citizen monitoring programs using acoustic data have been useful for detecting population and community patterns. However, they have rarely been used to study broad scale patterns of species traits. We assessed the potential of acoustic data to detect broad scale patterns in body size. We compared geographical patterns in body size with acoustic signals in the bat species *Pipistrellus pipistrellus*. Given the correlation between body size and acoustic characteristics, we expected to see similar results when analyzing the relationships of body size and acoustic signals with climatic variables.
**Methods**. We assessed body size using forearm length measurements of 1,359 bats, captured by mist nets in France. For acoustic analyses, we used an extensive dataset collected through the French citizen bat survey. We isolated each bat echolocation call ($n = 4,783$) and performed automatic measures of signals, including the frequency of the flattest part of the calls (characteristic frequency). We then examined the relationship between forearm length, characteristic frequencies, and two components resulting from principal component analysis for geographic (latitude, longitude) and climatic variables.
**Results**. Forearm length was positively correlated with higher precipitation, lower seasonality, and lower temperatures. Lower characteristic frequencies (i.e., larger body size) were mostly related to lower temperatures and northern latitudes. While conducted on different datasets, the two analyses provided congruent results.
**Discussion**. Acoustic data from citizen science programs can thus be useful for the detection of large-scale patterns in body size. This first analysis offers a new perspective for the use of large acoustic databases to explore biological patterns and to address both theoretical and applied questions.

Corresponding author
Caterina Penone,
caterina.penone@gmail.com

# INTRODUCTION

Acoustic methods can be used for biodiversity assessment (*Riede, 1998*; *Johnston et al., 2008*; *Newson, Evans & Gillings, 2015*) and a large amount of sound data is available in repositories, and through large-scale citizen science programs (*Walters et al., 2012*;

Jones et al., 2013; Penone et al., 2013; Newson et al., 2017). Improved computational facilities and new algorithms now enable acoustic data to be recorded and analyzed automatically to identify species and collect information on each recorded individual (Bardeli et al., 2010; Walters et al., 2012; Jones et al., 2013; Newson et al., 2017). Despite their high potential, acoustic databases have rarely been used to explore trait patterns across large spatial extents (Law, Reinhold & Pennay, 2002, but see Murray, Britzke & Robbins, 2001). Here, we examined whether acoustic data collected by a citizen science program might be used to detect broad scale patterns in species traits, focusing on body size.

Body size is closely related to many physiological, life history, and ecological parameters (Peters, 1986). At a macroecological scale, this trait is linked to species ranges, abundances, co-occurrences, and energy use (Brown, 1995; Barnagaud et al., 2014). Therefore, assessing broad scale patterns in body size can help in understanding how resources and diversity are partitioned in space and time, as well as life history evolution.

For a number of taxonomic groups, geographic variation in body size within species has been related to latitude and climatic variables such as temperature and precipitation (Bergmann's and James' rules: Bergmann, 1847; James, 1970). Organisms generally show increased body size or mass at higher latitudes, in colder and wetter climates, and where there is increased productivity or food availability (Rosenzweig, 1968; Ashton, Tracy & De Queiroz, 2000; Freckleton, Harvey & Pagel, 2003; Meiri & Dayan, 2003; Yom-Tov & Geffen, 2006; McNab, 2010; Clauss et al., 2013). Even if originally described among closely-related species of homeotherms, it is now recognized that such macroecological rules also apply at the intraspecific level (Ashton, Tracy & De Queiroz, 2000). At the intraspecific level, adult body size is also partly related to food availability during early-stage growth (Henry & Ulijaszek, 1996; Lindström, 1999). Here we analyzed the geographic variation in body size of Pipistrellus pipistrellus (Schreber, 1774), a common and sedentary bat species of the Vespertilionidae family, which is widely distributed and abundant in several European habitats, including France (Davidson-Watts, Walls & Jones, 2006; Mitchell-Jones et al., 1999).

The link between body size and climatic variables at the intraspecific level has already been highlighted in some Vespertilionidae species (e.g. Burnett, 1983; Ashton, Tracy & De Queiroz, 2000; Meiri & Dayan, 2003), but the validity of the rule for P. pipistrellus has not yet been demonstrated (Stebbings, 1973; Jones, Van Parijs & Vanparijs, 1993). For many echolocating bats, the peak frequency (i.e., the frequency of maximum energy), and the characteristic frequency (i.e., the frequency of the flattest part of the echolocation call, Kalko & Schnitzler, 1993), have been shown to be negatively correlated with body size (Barclay & Brigham, 1991; Jones, 1999; Jakobsen, Ratcliffe & Surlykke, 2013; Jung, Molinari & Kalko, 2014; Thiagavel, Santana & Ratcliffe, 2017). Even at the intraspecific level, smaller individuals emit signals of higher frequency than larger ones, a trend that has been mostly attributed to allometric constraints and to the size of the target, i.e., prey (Barclay & Brigham, 1991; Jones, 1999; Gillooly & Ophir, 2010; Thiagavel, Santana & Ratcliffe, 2017). Allometric constraints imply that larger bats produce lower frequency sounds because they have a bigger larynx and larger resonant chambers (Jacobs, Barclay & Walker, 2007). This

negative correlation between body size and characteristic frequency is particularly strong in the Vespertilionidae family (*Jones, 1999*; *Thiagavel, Santana & Ratcliffe, 2017*). However, this pattern is not valid for all bat species, such as most of the so-called constant frequency bats (*Russo, Jones & Mucedda, 2001*), because of important differences in the mechanisms involved in their echolocation modalities. *P. pipistrellus* does not belong to this group of species and is therefore a potentially interesting model species to explore biogeographic patterns, using characteristic frequencies as a proxy for body size.

To assess whether large-scale patterns in body size are detectable using bat echolocation frequencies, we (i) studied the geographic variation in body size of *P. pipistrellus* and its environmental correlates, using forearm length measurements, which are proportional to body size (*Thiagavel, Santana & Ratcliffe, 2017*). We then (ii) examined the same relationships using the characteristic frequency of *P. pipistrellus*. Given the negative relationship between characteristic frequencies and body size (*Thiagavel, Santana & Ratcliffe, 2017*), we expected to see similar relationships between forearm lengths and characteristic frequencies, and geographic and environmental variables.

## MATERIALS & METHODS

### Mist net dataset

In order to relate body size characteristics to geographic (latitude, longitude) and climatic variables for *P. pipistrellus* we assembled several datasets obtained from mist net bat captures and measurements in continental France. Data were collected by bat workers for local conservation studies and inventories from April to July over a 4 year period (2009–2012), representing 208 observations in 186 localities. Forearm length was measured with calipers and was used as a measure of body size. Forearm length is roughly proportional to the mass of the individual with a 0.3 ratio in Vespertilionids (*Norberg, 1981*; *Thiagavel, Santana & Ratcliffe, 2017*) and this was confirmed by our data (see Appendix A3). However, in contrast to body mass, forearm length is not subject to seasonal or reproductive changes and is therefore a more reliable proxy of body size (*Thiagavel, Santana & Ratcliffe, 2017*). Species identification, sex, reproductive status, and age estimation were assessed in the field according to *Dietz & Von Helversen (2004)*. We removed from our dataset juveniles and individuals for which we did not have sex information. Bats were captured by bat workers under license from the French Ministry of Environment. All captures were made according to the code of ethics drafted as part of the national training system for the capture of bats (*MNHN & SFEPM, 2015*). Individuals were released immediately after they had been measured; therefore, the acoustic parameters of the captured individuals are not available for this dataset. This is a limitation of our study; however, obtaining frequency information for each captured individual would have been challenging because such recordings have to be done under laboratory conditions given that hand-released bats usually provide atypical calls (*Britzke, 2004*).

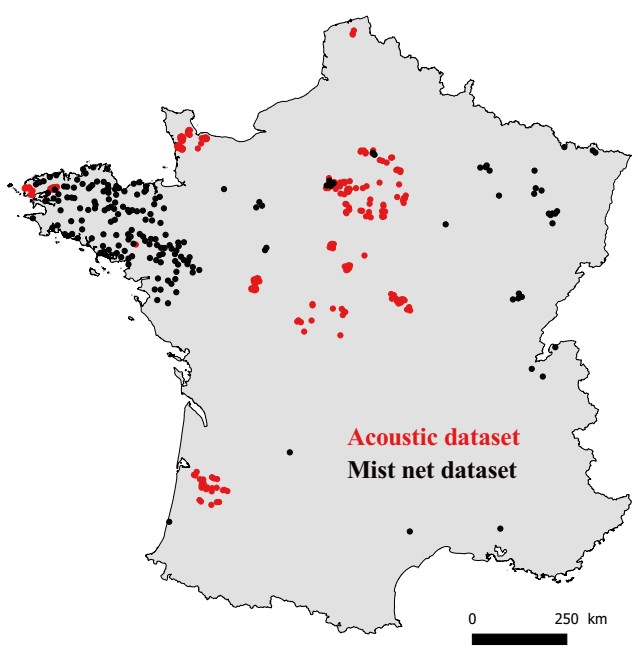

**Figure 1**  Map of France with locations where bats were captured (mist net dataset, black points), and recorded (acoustic dataset, red points).

## Acoustic dataset and sound analyses

To examine the relationship between acoustic data and geographic and climatic variables we used a dataset provided by the French national bat monitoring program (*Vigie-Chiro*), a citizen science program coordinated by the French National Museum of Natural History (*Kerbiriou et al., 2018*, Appendix A1). Observers completed two surveys in June to July and August to September 2009, along the same 30-km circuits. Each circuit was divided into ten 2-km road segments, separated by 1-km road segments in which no records were made.

We analyzed the results from 76 of these 30-km circuits, for which we had complete data on each variable (see variable descriptions below and Fig. 1).

We used Scan'R software (version 1.5; Binary Acoustic Technology, Tucson, AZ, USA) to isolate each *P. pipistrellus* call and perform automatic extraction of signal parameters (Appendix A1). Very low quality calls were automatically filtered out by Scan'R, which was set up to remove false detections due to noises such as rain drops or static discharging ("pop filter", in Scan'R manual, *Binary Acoustic Technology LLC, 2009*). Furthermore, we retained only signals longer than 1 ms and set the "fit restricted" parameter to "medium" (i.e., all detections are required to match the internal curve model, with some tolerance for error, see Scan'R manual; *Binary Acoustic Technology LLC, 2009*). A manual inspection of around 50 randomly chosen signals confirmed us that they were indeed from *P. pipistrellus*. Analyses were performed using the LowFc (hereafter "characteristic frequency"), which is the frequency at the end of the shallowest modulated part of the calls (*Kalko & Schnitzler, 1993*; Fig. A2.1). In order to avoid potential misidentification of *P.pipistrellus* we selected only calls with a LowFc between 41 and 52 kHz (*Barataud et al., 2015*; Appendix A2). We
also extracted the slope of the quasi-constant frequency (QCF) characteristic part of the acoustic signal ("SlopeQCF", Fig. A2.1). This value has been shown to be correlated to frequency, and is higher (i.e., more negative) in cluttered habitats (i.e., habitats with closed and dense vegetation) compared to open habitats, including for *P. pipistrellus* (*Kalko & Schnitzler, 1993*). This means that bats emit higher calls with a steeper slope (i.e., higher values of SlopeQCF) when flying in cluttered habitats (*Kalko & Schnitzler, 1993*). In open habitats, their calls have a much shallower frequency modulation (corresponding to low values of SlopeQCF) and lower frequencies (*Kalko & Schnitzler, 1993*). This was verified by our data (see Appendix A2 for a detailed description). Studies considering geographical variation in echolocation calls might be affected by the clutter conditions in the sample areas (*O'Farrell, Corben & Gannon, 2000*). We thus included SlopeQCF in our models, to take into account the effect of habitat clutter at a local scale on the characteristics of echolocation signals.

## Geographic and climatic variables

Latitude has been used as a proxy for temperature in several biogeography studies (e.g. *Ashton, Tracy & De Queiroz, 2000*). However, climatic variables other than temperature (e.g., rainfall) covary with latitude and might be related to body size. We extracted data on climatic variables for both mist net points and acoustic data segments from WorldClim 1.4 (*Hijmans et al., 2005*). We selected the annual mean temperature (Bio1), annual temperature range (Bio7, an index of seasonality), mean temperature of the warmest and coldest quarters (Bio10 and Bio11, respectively), annual precipitation (Bio12), and precipitation of the wettest quarter (Bio16). Because body size patterns might reflect climatic variations at large temporal scales (*Brown & Maurer, 1989*), we used averaged WorldClim variables across 30 years (1960–1990). Although this information does not overlap our recording and capture periods, it is usually considered as representative of current climatic conditions in several studies (e.g. *Machac et al., 2011*; *Hassall et al., 2014*). Since climatic variables tend to covary in space, we used a principal component analysis (PCA) to combine the information on geographic (latitude and longitude) and climatic variables. All variables were scaled prior to PCA. We also measured the mean altitude in each segment and mist net point, using a digital elevation model at 250 m resolution (BD-Alti IGN).

## Statistical analyses

Altitude might have an effect on body size (*Meiri & Dayan, 2003*); therefore, we restricted the analysis to points below 500 m and removed island data to avoid insularity issues. Since data collected before 2009 might contain spurious identifications owing to confusion with *P. pygmaeus* (*Mayer & Von Helversen, 2001*), we considered only data collected after this date, for which identification of *P. pipistrellus* was reliable.

We fitted linear mixed-effects models (lme4, lmerTest (*Bates et al., 2015*; *Kuznetsova, Brockhoff & Christensen, 2017*) in R (*R Core Team, 2013*) to determine if forearm length and characteristic frequencies (response variables) were related to latitude and large-scale climatic variables, represented by principal components. In the model with mist net data,

we corrected for the day of the year in which the bats were captured, and for the sex of the individual by adding these variables as covariates (see Appendix A3 for more details on the effect of sex). We included the name of the locality where the individuals were captured, nested into the year of capture as random effect, to account for temporal and spatial structure of the data. Given the high concentration of data points in the northwestern part of France, in order to check for potential bias in our analysis, we repeated the model on a subset which included only the northwestern points (see Appendix A4 for more details).

In the model with acoustic data, we corrected for the survey period (first/second), and for the slope of the quasi-constant frequency (SlopeQCF) of each detected acoustic signal. Given that several bat species, including *P. pipistrellus*, tend to modify their calls in order to avoid interference with calls of conspecifics (also called jamming avoidance, *Necknig & Zahn, 2011*; *Ulanovsky et al., 2004*), we incorporated the number of calls within a 400 m section into the models (call abundance), as a proxy for the number of individuals. Segments nested into each circuit were included as a random effect, to take into account the spatial structure of the data. The local temperature during recording sessions might have influenced call detection; however, we only had information on this parameter for 44 circuits out of 76. Therefore, we ran the analysis on these 44 circuits ($n = 1,705$ calls), in order to assess the importance of temperature during recording sessions, and its potential bias on our results (Appendix A5). The day of the year in which the bats were recorded could potentially be more informative than the survey period, but we had information for 45% of the data points; we thus tested the effect of the date of recording on this data subset. Given the high concentration of data points in the northern part of France, we repeated the model on a subset including only the central northern points (Appendix A4). We also ran the analyses on the southern points only, to check for potential species identification issues (further details in Appendix A4).

Finally, given that females tend to be bigger than males (Appendix A3), variation in the sex ratio across the climatic gradient might affect the results for the acoustic dataset for which we do not have information on the sex of individuals. Therefore, we tested the relationship between the sex-ratio and climatic variables using the mist-net dataset. We visually checked all models for normality in model residuals and heteroscedasticity (see R script: https://github.com/caterinap/pipip_frequency_bodysz).

## RESULTS

We analyzed mist net data for 1,359 individuals. Forearm length of *P. pipistrellus* varied between 27.6 and 37.4 mm (mean $31.6 \pm 0.96$ SD). The number of individuals captured per location and per year varied between one and 52 (mean $6.5 \pm 6.1$ SD).

Along the 69 circuits with available acoustic and environmental data, we analyzed 4,783 calls of *P. pipistrellus*. The characteristic frequency varied between 42.1 and 51.9 KHz (mean $46.9 \pm 1.9$ SD). The number of recorded calls per segment varied between 1 and 71 (mean $23.5 \pm 17.7$ SD).

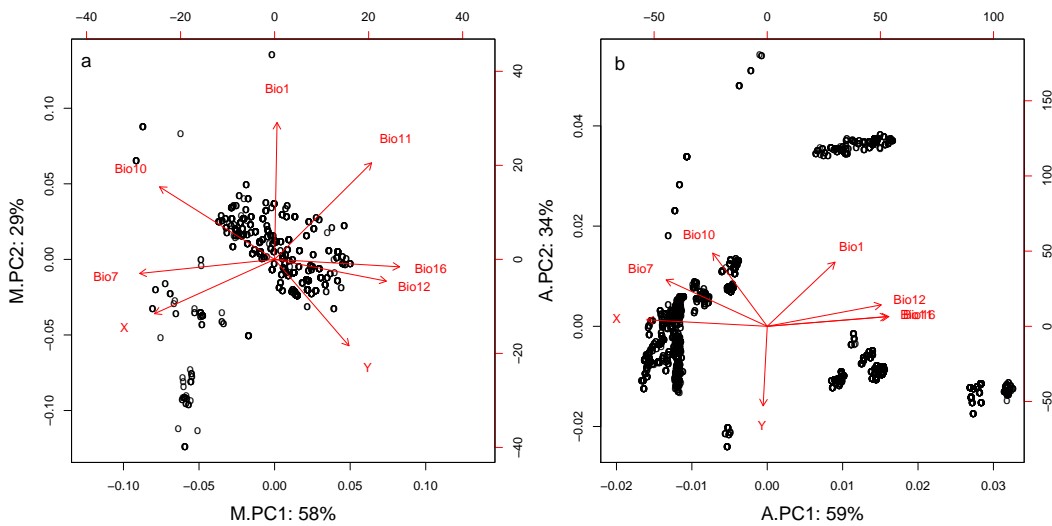

**Figure 2** First two principal components resulting from PCA for geographic and climatic variables for mist net (A) and acoustic (B) datasets. Percentage values represent proportion of the total variation explained by each component. For variables descriptions see Table 1.

## Geographic and climatic variables

The two first principal components of the PCA for both the mist net (M.PC1 and M.PC2) and the acoustic (A.PC1 and A.PC2) datasets summarized 87% and 93% of the variation in geographic and climatic variables, respectively (Fig. 2 and Table 1). As shown by variable loadings for the principal components (Fig. 2 and Table 1), M.PC1 and A.PC1 mostly included information about longitude, temperature variability, and precipitation in both datasets. High M.PC1 or A.PC1 values represented places with high precipitation (Bio12, Bio16), and low temperature range or seasonality (Bio7), principally located in western France. The secondary principal component was more representative of latitude (especially for the acoustic dataset, A.PC2) and temperature (Fig. 2). High A.PC2 or M.PC2 values characterized places with high temperatures in both the warmest and coldest quarters of the year (Bio1, Bio10, and Bio11), and were principally located in southern France. However, for the acoustic dataset, the PCA clearly discriminated between longitudes (for A.PC1) and latitudes (for A.PC2). In contrast, for the mist net dataset, both M.PC1 and M.PC2 carried information about both latitude and longitude, with high M.PC1 values representing places with low temperatures (Bio10, Bio11) located in northern France. This is likely linked to the fact that the capture points are more spread along an east–west axis while the acoustic circuits are more distributed along a north-south axis.

## Mist net dataset

Using mist net capture data, we found a significant positive relationship between body size (represented by forearm length) and the first principal component (M.PC1) (Table 2, Fig. 3). Therefore, larger individuals were captured at places characterized by high precipitation (Bio12, Bio16), and a low temperature range (Bio7), principally located in northwestern France. We also found that males were smaller than females (negative

**Table 1  Variable loadings resulting from principal components analysis of geographic and climatic variables for the mist net and acoustic datasets.** Only the first two PCs are shown and were used for further analysis.

|  | | Mist-net dataset | | Acoustic dataset | |
|---|---|---|---|---|---|
|  | Description | M.PC1 | M.PC2 | A.PC1 | A.PC2 |
| X | Longitude | −0.40 | −0.26 | −0.44 | 0.05 |
| Y | Latitude | 0.25 | −0.41 | −0.01 | −0.58 |
| Bio1 | Annual mean temperature | 0.01 | 0.65 | 0.25 | 0.47 |
| Bio7 | Annual temperature range | −0.45 | −0.07 | −0.37 | 0.34 |
| Bio10 | Mean temperature of the warmest quarter | −0.39 | 0.34 | −0.20 | 0.54 |
| Bio11 | Mean temperature of the coldest quarter | 0.33 | 0.46 | 0.44 | 0.07 |
| Bio12 | Annual precipitation | 0.37 | −0.10 | 0.42 | 0.16 |
| Bio16 | Precipitation of the wettest quarter | 0.42 | −0.04 | 0.45 | 0.07 |

**Table 2  Results of the fixed effects from linear mixed-effects models for the relationships between forearm length (from mist net data), characteristic frequencies (from acoustic data) and climatic and geographic variables represented by principal components.**

|  | Estimate | SE | $t$-value | P |
|---|---|---|---|---|
| **Forearm length ($n = 1351$)** | | | | |
| Male | −0.76 | 0.05 | −15.74 | <0.000 |
| Day of the year | −0.00 | 0.00 | −0.33 | 0.744 |
| M.PC1 | 0.05 | 0.01 | 3.42 | <0.000 |
| M.PC2 | −0.02 | 0.02 | −0.87 | 0.385 |
| **Characteristic frequency ($n = 4783$)** | | | | |
| SlopeQCF | −0.72 | 0.04 | −27.62 | <0.000 |
| Second Survey | 0.68 | 0.06 | 11.76 | <0.000 |
| Call abundance | −0.00 | 0.00 | −0.25 | 0.806 |
| A.PC1 | −0.00 | 0.03 | −0.13 | 0.900 |
| A.PC2 | 0.11 | 0.03 | 3.10 | 0.003 |

Notes.
SlopeQCF, slope of the quasi-constant-frequency.

intercept, Table 2). This result is expected since this species exhibits a sexual dimorphism in body size (see Appendix A3 and (*Dietz, Von Helversen & Nill, 2007*)). However, we did not find a significant relationship between the sex-ratio and M.PC1 or M.PC2 (estimates: 0.01 ± 0.04 and 0.02 ± 0.05, respectively and both $p > 0.7$), therefore changes in sex ratio should not affect our main conclusions. Finally, we did not find an effect of the date of measurement on body size; however, this was expected given that we analyzed only adults, and corrected for the sex of the individuals.

## Acoustic dataset

Using acoustic data, we found a significant positive relationship between characteristic frequencies and the second principal component (A.PC2) (Table 2, Fig. 3). Therefore, lower frequencies (i.e., larger individuals (*Barclay & Brigham, 1991*; *Jones, 1999*; *Jakobsen, Ratcliffe & Surlykke, 2013*; *Jung, Molinari & Kalko, 2014*)) were related to places characterized by
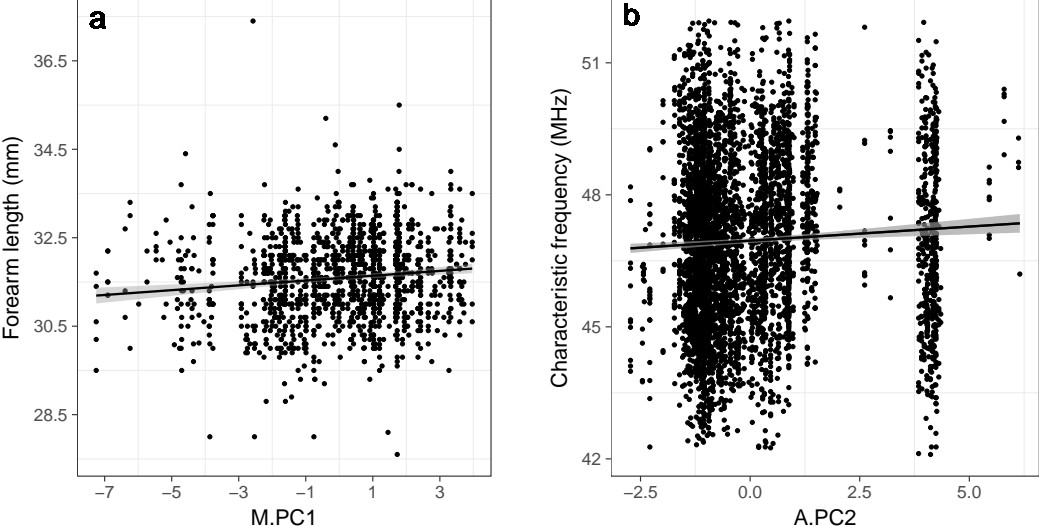

**Figure 3  Relationships between forearm length (A) and characteristic frequency (B) and the principal components resulting from PCA (PC1 and PC2) that were found to be significant in linear mixed-effects models.** The lines represent fitted linear models with 95% confidence region for the regression fit.

low temperatures (Bio1 and Bio10) and a low temperature range (Bio7), principally located in northern France. The models showed significant effects of the survey period and SlopeQCF. Characteristic frequencies were higher when measured during the second survey period (August–September). Characteristic frequencies were also higher for higher absolute values of SlopeQCF, i.e., for steeper signals, harboring a more negative slope (Table 2). SlopeQCF was higher in cluttered habitats (Appendix A2). When the local temperature during a recording session or the call abundance were not included in the model, the results remained quantitatively comparable (Table A5.1). We also obtained similar results when using the date of recording instead of the survey period in the models (Appendix A6). Neither our capture localities nor acoustic circuits were evenly distributed in space. However, the main results were maintained when the analyses were conducted on spatial data subsets with high point densities (Appendix A4), and they were similar for both mist net capture and acoustic datasets, thus suggesting that our overall results were robust.

## DISCUSSION

Using acoustic data from a large-scale citizen science program, we found significant relationships between characteristic frequencies of *P. pipistrellus* and geographic and climatic variables. We detected lower frequencies, corresponding to larger body sizes, in colder regions with low temperature variation. In parallel, our analysis using measurements of bat body size in the field showed that larger bats were also captured in colder and wetter areas, with low temperature variation. Our datasets did not cover the same regions: the

acoustic dataset spans a latitudinal gradient, while the mist-net dataset rather includes a longitudinal sampling. Despite these differences, the two analyses showed similar results, which indicates that acoustic datasets might be used to study large-scale patterns in body size.

Forearm length was mainly related to high precipitation, low seasonality, and low temperatures occurring in sample areas in northwestern France. Lower characteristic frequencies (i.e., larger body sizes) were mostly correlated with the low temperatures that occur in northern France. Our result is in accordance with a previous study on *P. pipistrellus* conducted over a similar spatial extent to the present one (*Stebbings, 1973*). Furthermore, this pattern of body size increasing with decreasing temperature and increasing humidity has also been found for *Eptesicus fuscus*, a bat species from temperate zones (*Burnett, 1983*), and for *Cynopterus sphinx,* a tropical bat species (*Storz et al., 2001*). More importantly, our results are also in accordance with biogeographic rules, that predict increased body sizes at higher latitudes and in colder and wetter climates (Bergmann's and James' rules: *Bergmann, 1847*; *James, 1970*).

Intraspecific variation in body size across a species range might be driven by several mechanisms including adaptation to local climate (*Endler, 1977*; *Cordero & Epps, 2012*) and adaptive plasticity as a response to different environments and to resource availability during early development (*Henry & Ulijaszek, 1996*; *Lindström, 1999*; *McNab, 2010*). At this stage, the drivers of any spatial patterns observed remain to be identified. While *P. pipistrellus* is a mobile species with a broad and continuous distribution, a genetic structuring hypothesis cannot be ruled out because such structuration has been observed within similar species (*Kerth & Petit, 2005*; *Moussy et al., 2013*). A non exclusive alternative hypothesis to explain the observed pattern could be the spatial difference in resource availability during the early stages of development. Future research using genetic information and data on resource availability could help to assess the relative importance of different drivers of body size variation in *P. pipistrellus.*

We also found other interesting results that reinforced the main ones and further suggested the potential of acoustic monitoring for the detection of body size patterns at large spatial scales. For instance, characteristic frequencies during the second survey, i.e., in August-September, were higher than in the first one. If frequency is linked with body size as expected (*Thiagavel, Santana & Ratcliffe, 2017*), this pattern would indicate a higher proportion of smaller individuals in the population during this period. This might be explained either by more intense male activity (as they are smaller than the females), or by a higher proportion of young bats (which are smaller than the adults) in August to September (*Arthur & Lemaire, 2015*). We also found that SlopeQCF was positively related to the characteristic frequency and was higher in cluttered habitats; a result that has previously been described at the individual level (*Kalko & Schnitzler, 1993*). Therefore, broad scale acoustic monitoring may also have the potentiality to enable the detection of fine-scale behavioral adaptations to the local environment, such as vegetation clutter.

In France, other species also emit calls with terminal frequencies between 41–52 kHz (*P. nathusii, P. kuhlii, P. pygmaeus,* and *Miniopterus schreibersi*). Therefore, some identification errors could have occurred due to overlap between acoustic repertoires. However, our dataset contained few samples from the Mediterranean region; thus,

the abundance of *P. pygmaeus* and *M. schreibersi*, which are mostly present in the Mediterranean (*Arthur & Lemaire, 2015*) should be very low in our dataset. *P. kuhlii* has lower terminal frequencies on average than *P. pipistrellus* (*Obrist, Boesch & Flückiger, 2004*; *Barataud et al., 2015*), but is much more abundant in the south than in the north of France (*Arthur & Lemaire, 2015*), which would contradict the observed pattern (larger individuals and lower calls in northern France). Therefore, even allowing for errors in identification, potential confusion with other species should not significantly bias our results.

## CONCLUSION

Using bat characteristic frequencies collected by a citizen science program, we identified apparent intraspecific large-scale variation in body size. This result was also supported by mist net capture data and highlights new perspectives for the use of acoustic databases to explore biological patterns at various spatial and temporal scales. In this context, characteristic frequencies can be used as a proxy for body size, and because body size is also related to habitat suitability and environmental filtering (*Cisneros et al., 2014*; *Nash et al., 2014*), they might constitute useful indicators for the assessment of biodiversity states and trends. Further studies might cover a larger geographic range focus on other bat species and explore deeply all the potentialities of characteristic frequency measurements to address both fundamental questions, such as the validity of the Bergmann rule, and applied ecological questions, such as the relationship between habitat suitability or degradation and body size. Large-scale data from citizen science programs might also benefit research projects that aim to assess the mechanisms underlying biodiversity responses to global changes at the intraspecific level (*Luo et al., 2014*), an urgent topic in the context of the current biodiversity crisis.

## ACKNOWLEDGEMENTS

The success of such long-term, large-scale surveys relies entirely on the continuous participation of voluntary observers that we warmly thank. We are also deeply grateful to bat-workers for providing mist net capture data. Names of bat-workers and volunteers are cited in the Supplementary material of this paper Appendix A7, extended acknowledgements. We also thank Robin Julien for help in analyzing acoustic data and two anonymous reviewers for helpful comments and suggestions.

### Funding

The authors received no funding for this work.

### Competing Interests

The authors declare there are no competing interests.

## Author Contributions

- Caterina Penone and Christian Kerbiriou conceived and designed the experiments, performed the experiments, analyzed the data, contributed reagents/materials/analysis tools, prepared figures and/or tables, authored or reviewed drafts of the paper, approved the final draft.
- Jean-François Julien and Julie Marmet conceived and designed the experiments, performed the experiments, contributed reagents/materials/analysis tools, authored or reviewed drafts of the paper, approved the final draft.
- Isabelle Le Viol conceived and designed the experiments, performed the experiments, analyzed the data, contributed reagents/materials/analysis tools, authored or reviewed drafts of the paper, approved the final draft.

## Animal Ethics

The following information was supplied relating to ethical approvals (i.e., approving body and any reference numbers):

Bats were captured by bat workers under license from the French Ministry of Environment.

## Data Availability

The data is available in the Supplemental File and the code is available at GitHub: https://github.com/caterinap/pipip_frequency_bodysz/blob/master/Stats_plots_pipip. R.

## Supplemental Information

Supplemental information for this article can be found online at http://dx.doi.org/10.7717/peerj.5370#supplemental-information.

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
