# Peer review of "Body size information in large-scale acoustic bat databases"

_PeerJ, doi:10.7717/peerj.5370_

## Round 0.1 · original submission · Major Revisions

The reviewers and I agree that this paper is a valuable addition to the literature on bat ecology. The manuscript would benefit from a thorough proofreading from a native English speaker. Please also address each of the points raised by the reviewers. Also ensure that the supplementary materials are in English and are formatted for ease of use.

Reviewer 1 ·

Basic reporting

Clear language:
I had edit and re-read the manuscript several times to understand the material as written. The language is often unsophisticated or casual. Moreover, preparation appears rushed (especially the Supplemental Materials), as formatting, names, and style are inconsistent and difficult to follow. I urge the authors to consult a native English speaker to increase the quality of the manuscript.

The intro and background:
The use of a citizen-science project seems central to the manuscript topic, but the authors fail to demonstrate that the quality of the data is appropriate for the study of bat body size measurements and acoustic call variation.
I'm unclear as to the most important broader implication of this research as written. What's the central idea? Is it about the applicability of citizen science for acoustic data collection? Or that both bat calls and body size show a consistent trend in relation to geography and climate, making calls an appropriate proxy for body size in future studies?
I expected more detailed background on P. pip or relevant conspecifics given that a single species, in particular, was featured in the research. Also, why was this species a model system for this research over any other species? Why is France appropriate for demonstrating variation in climate and geography?

The referenced literature:
Only two cited papers were published post-2015, suggesting that the authors have not updated the manuscript references to account for recent findings.
Citations are sometimes not provided where needed (e.g., lines 55, 81, 106, 173, etc.).

The structure:
Overall, I recommend restructuring the introduction to clarify the aims, questions, and reasoning for the study. In particular, the final paragraph could be more succinct, and the hypotheses made clearer.
The methods, especially, could have been easier to follow if consistent identifiers were used throughout. For instance, reference to the French citizen bat survey changes throughout when it could have been referred to as Vigie-chiro (which is named only in the abstract). Is it the "mist-net" dataset or "capture" dataset? Moreover, although I understand the difference between Fc and SlopeQCF, they are hard to keep track of because they are not consistently named. I believe that more attention should be given to the writing to ensure consistent language.
Since there are two PCA results reported, I suggest naming the axes with respect to the dataset for clarification when reading. I suggest looking at other papers describing results of PCA for tips on language usage.

Figures and tables:
Maps of the study locations should be provided in the main manuscript. This can be used to show the trends in important data. For instance, sampling points could use different sized icons to illustrate larger or smaller values.
Since PCA was a major analysis for the study, I suggest a table of each variable and its corresponding eigenvectors for the two important axes.

Raw data:
Supplied, though some columns are written in French.

Experimental design

The topic:
Is original and within the scope of PeerJ

The research questions:
I recommend using numbers to distinguish between each question/aim and hypothesis in the order of how they will inform one another.

Technical and ethical standard:
I did not see permitting information or ethical standards reported in the main manuscript.

Methods:
I do not believe the methods in the main manuscript is clear without referring to the Supplemental Materials.
Sample sizes and descriptions of each variable should be provided in a table within the main manuscript.
"Clutter" must be defined.
The authors did not report their settings for acoustic analysis they used to filter calls. They also did not report how they chose the quality of calls to retain for analyses, nor how many calls per recording they used. Did they average the 10 highest quality calls per recording? Per sampling segment? The sampling protocol, processing, and assumptions for acoustic recordings are crucial for the validity of the findings.
For the analyses of bat calls, I was not clear on the sample size after re-reading. If a single sampling circuit (n=78) was cut into segments (n= 1133) and used for acoustic analyses, thus increasing the sample size by non-independent data points, this would lead to pseudo-replicated.
I was unclear on how climatic data was acquired and whether temporal and spatial autocorrelation was addressed. For instance, climatic variables for more than just the season of capture would matter for body traits because climate has affected the population over time.
How do differences in bat calls by sex affect the acoustic results given that the authors state call frequency varies by sex?

Validity of the findings

Replication:
I question is the acoustic data was pseudo replicated if sampling circuits were split into multiple segments for analyses outside of averaging bat call parameters (which is unclear as written).

Data usage:
Given that body size is the typical variable used in these types of studies, I was surprised that the authors did not show evidence from their dataset that body size is consistently correlated with forearm length across the sampling area. If the data were available, why not make the analyses more robust by showing the correlations between multiple body trait measurements?
How representative are the measurements made for P. pip of variation across France? The spatial distribution of the data point were not similar between the two datasets that were compared. Moreover, within a dataset it is clear that the sampling was not random nor distributed geographically. Given that this is a study looking at geographic and climatic variation, I would have expected a more representative sampling distribution.

Conclusions:
What is the relationship between forearm length, geography, and climate for P. pip? (Rules have not been verified for this species.)
I am not comfortable with the conclusion that a single parameter of bat acoustic calls can be used as proxy for forearm length given the analyses performed here. The differences in the sampling distribution of the two compared datasets is massive and, as stated by the authors, are not distributed across the geographic or climatic ranges of France.

Speculation:
There should be more discussion about the validity of Bergmann's and James' rules given their relevance to the research.

Additional comments

I had to edit and re-read the manuscript several times to understand the material as written, and I do not believe the main manuscript is clear without referring to the Supplemental Materials frequently. The writing is often unsophisticated and casual. Moreover, preparation appears rushed (especially the Supplemental Materials), as formatting, names, and language are inconsistent and difficult to follow. I urge the authors to consult a native English speaker to increase the quality of the manuscript.

Reviewer 2 ·

Basic reporting

The standards of the journal have generally been met in the manuscript, with a clear and professional structure including stated goals and background, methods, results, and discussion that pertain to the stated goals. That said, I failed to find information regarding the ethical use of animals and some minor errors to note:
1. Line 12 - I believe this should be Sorbonne (not Sorbone)
2. Line 44 - 'calculation algorithms' is redundant - recommend deleting 'calculation'
3. Lines 62 and 67- At ^the intraspecific level
4. In all materials (manuscript + supplementary), please be sure to define your acronyms before you use them for the first time - e.g., QCF, MNHN
5. Style point - e.g. and i.e. are typically followed by a comma
6. Line 201 and other - 'precipitations' should be 'precipitation'
7. Line 240 [This pattern of body size]……has
8. Line 261 - may also ^have the potentiality to allow detecting fine scale….
9. Line 285 - 'form' should be 'from'
10. Line 286 - 'that aim ^to at assess…'

I do, however, recommend that a greater attention to detail, proper English spelling and grammar is given to the supplementary material and data files. For example, in Appendix I - page 4 - this sentence is repeated 'The results of the cross-validation on more that 58% of the data indicate that for 8% of the calls identified as P. kuhlii it was not possible to refute potential contact of P. nathusius and among the P. pipistrellus less than 0.1% could be P. nathusii'. Also, Figure A2.1 legend needs to reference panel A correctly.

Specific comments for the data files:
1. description of field names (headings) should include units
2. the wind speed field consist of French words describing conditions - an English translation of each of the words in the description of the field would be helpful
3. Is time provided in UTC?

Experimental design

This manuscript takes advantage of the established relationship between body size and call frequencies in bats to apply the acoustic data collected through a citizen science bat acoustic monitoring project throughout France to a relevant ecological question about whether intraspecific variation in body size varies systematically with climate. Given that this is the first study (of which I and the authors are aware) that attempts this approach, the authors endeavor to confirm the validity of the findings against a comparable analysis using actual body size data collected from bat capture efforts from throughout France. Both sets of data appear to be robust and collected over the same time period and spatial range. However, the distribution of points does vary significantly between the datasets, with the capture dataset largely from northwestern France and the acoustic dataset largely from central-northern France. The geographic biases in the datasets are the greatest weakness in the study, in my opinion, but do not negate the findings. Bat data are extremely challenging to collect, capture data in particular, and thus sparsity and patchiness of data availability are the norm for ecological studies involving bats. Studies such as this that evaluate the potential diverse applications of the (relatively) more easily acquired acoustic data are therefore of great value to the study of bat ecology.

Manuscript appears to have been prepared to journal standards, with some minor comments:

1. Line 96 - Please clarify what is intended by the phrase 'capture points'. I presume that refers to unique combinations of data and site, but it is not clear. How many sites across how many degrees of latitude were included? I recommend that Figure A1.1 be included in the main manuscript.

2. Please provide more details about the WorldClim analysis - especially, how many years of data were used? Restricted to time period of capture and acoustical data would be the most compelling, of course.

3. Please provide more details about the suitability of the data for the linear mixed effects model approach, including any evaluation of the variance, independence, and distribution of the errors.

Validity of the findings

In this study, the authors apply traditional principal components techniques to reduce a diversity of relevant geographic and climate parameters into two principal components variables. These are then incorporated into linear mixed-effects models to evaluate whether actual measures of body size (empirical forearm lengths) and an acoustic body size correlate (characteristic low frequency) from Pipistrellus pipistrellus respond in similar ways to the geographic and climate principal components, while evaluating or incorporating the potential impacts of sex, temperature, year, and sampling location on the response. Significant, consistent relationships between body size and both measures of body size are found, although the slopes of the relationships are not striking. The authors thoroughly explore the potential biases in these findings in their analyses, as captured in the supplementary material, and they sufficiently explore the implications of their findings in the discussion. In short, their results suggest that the acoustic body size correlate is a valid proxy when actual body size data are unavailable.

Related comments:
1. I am struggling with the evaluation of the geographic biases in the data section of the Appendix and how it is conveyed in the manuscript. I appreciate that the authors are transparent about the geographic biases in the data, and I appreciate also that these are often unavoidable. I am concerned that merely splitting the data into two subsets and comparing results does not control for the effects of widely varying sample sizes. Subsampling of the dense data areas and/or evaluating the data subsets against the assumptions of linear mixed effects models would likely help to address this concern. Alternatively, acknowledging the bias and avoiding the subsetting analyses altogether would be acceptable from my perspective as well. If the spatial subset analysis is retained, I would consider rephrasing the concluding sentence capture din Lines 226-228. I do not agree that the 'main results were maintained when the analyses were conducted on spatial data subsets'. For example, the forearm length-climate relationship was nonexistent in the eastern dataset. Again, I think this could merely be insufficient data and therefore not readily interpretable or able to negate the main findings of this study.

2. As the authors note, the likely increased abundance of juveniles in the second acoustic survey could alter the intraspecific variation of body sizes and call frequencies. I recommend only using the first survey in the analysis, since the capture data did not include forearm lengths from juveniles.

Additional comments

I enjoyed this manuscript and, with the above comments and a few below addressed, believe it could be an important contribution to bat ecology, as the abundance of acoustic data increases significantly year over year.

A few minor comments:
1. Line 41 - I would argue that acoustic methods have already demonstrated their utility for biodiversity assessment, so recommend deleting 'promising'
2. Line 45 - ability to automatically identify species varies across taxonomic groups - recommend rephrasing to reflect the significant amount of research still remaining to be done in this area for many species
3. Line 53 - Brown's 1995 Macroecology book is a more appropriate reference for the sentence than Gaston & Blackburn 1996/Pyron 1999)
4. To further bolster the strength of the relationship between call frequency and body size, I recommend adding in a reference to Body Size Predicts Echolocation Call Peak Frequency Better than Gape Height in Vespertilionid Bats. Jeneni Thiagavel, Sharlene E. Santana & John M. Ratcliffe Scientific Reports 7, 828 (2017) doi:10.1038/s41598-017-00959-2

---

## Round 0.2 · Minor Revisions

The authors efforts in the revision process are greatly appreciated and have resulted in a robust manuscript. Thank you!

The submission is now scientifically acceptable, however there are a reasonable number of suggested edits from Reviewer 1 therefore we are returning the submission to you to give you a chance to incorporate any of these edits before final acceptance.

Reviewer 1 ·

Basic reporting

Generally, I think that this is a well-written manuscript with interesting implications. I believe that they find support for their hypothesis: they do show that acoustic data collected by a citizen science program can be used to detect broad scale patterns in species traits.

Please see the annotated copy of the manuscript to see where I made some minor changes and where I made comments.

Experimental design

The methods and results are much easier to follow. One thing I noticed this time is that the climate data overlaps with the sampling data for one year only, which might play a role in the strength of their findings. Also, there is only one year of overlap between the mist net and acoustic datasets, which might also introduce inconsistencies.

Validity of the findings

I believe that the authors must be more explicit about the limitations of the study given that the sampling locations are so different. Mist net results are most applicable to longitudinal sampling and acoustic results are most applicable to latitudinal sampling.

Also, I feel that they inflate the strength of the correlation between the two datasets. According to my interpretation of their results, although the first two PCA axes show similar loadings between the mist net dataset and acoustic dataset, the PCA axes that are significantly related to their two response variables are not the same. That is, PCA axis 1 is significant for forearm length, but PCA axis 2 is significant for frequency. Thus, the strongest climatic variables that concur between the two models are:
(1) large forearm/large body & low frequency/large body both relate to low temp range/low variation in temp, but stronger for large forearm, and
(2) large forearm/large body & low frequency/large body both relate to low temp in summer, but stronger for low frequency.

Additional comments

Thank you very much for addressing all of my recommendations. Again, please see the annotated copy of the manuscript.

Annotated reviews are not available for download in order to protect the identity of reviewers who chose to remain anonymous.

Reviewer 2 ·

Basic reporting

I appreciate the authors' efforts to improve the quality and clarity of the writing in this revision. I also appreciate that the authors incorporated my comments, regarding literature cited, figure selection, and raw data improvements.

Experimental design

I appreciate the authors additional efforts in response to the reviewers to evaluate the dataset for appropriateness for the model used and to further evaluate potential biases in the data.

Validity of the findings

Manuscript is much clearer and stronger in its intent and results, and I think will be a valuable contribution to acoustic ecology.

Additional comments

The authors efforts in the revision process are greatly appreciated and have resulted in a robust manuscript.

---

## Round 0.3 · accepted · Accept

I appreciate your effort to revise and improve this manuscript. I believe it is ready for release.

#